# A Machine-Learning Approach to Distinguish Passengers and Drivers Reading While Driving

**DOI:** 10.3390/s19143174

**Published:** 2019-07-19

**Authors:** Renato Torres, Orlando Ohashi, Gustavo Pessin

**Affiliations:** 1Institute of Exact and Natural Sciences, Federal University of Pará (UFPA), Belém 66-075-110 PA, Brazil; 2Informatics Department, Federal Institute of Pará, Vigia 68-780-000 PA, Brazil; 3Cyberspace Institute, Federal Rural University of Amazônia, Belém 66-077-830 PA, Brazil; 4Robotics Lab, Instituto Tecnológico Vale, Ouro Preto 35-400-000 MG, Brazil

**Keywords:** driver distraction, reading while driving, machine learning, deep learning

## Abstract

Driver distraction is one of the major causes of traffic accidents. In recent years, given the advance in connectivity and social networks, the use of smartphones while driving has become more frequent and a serious problem for safety. Texting, calling, and reading while driving are types of distractions caused by the use of smartphones. In this paper, we propose a non-intrusive technique that uses only data from smartphone sensors and machine learning to automatically distinguish between drivers and passengers while reading a message in a vehicle. We model and evaluate seven cutting-edge machine-learning techniques in different scenarios. The Convolutional Neural Network and Gradient Boosting were the models with the best results in our experiments. Results show accuracy, precision, recall, F1-score, and kappa metrics superior to 0.95.

## 1. Introduction

According to the World Health Organization, the number of road traffic deaths continues to increase, reaching 1.35 million in 2016 [1]. Road traffic injury is now the leading cause of death for children and young adults aged 5–29 years. As observed by Lipovac et al. [2], research conducted around traffic safety indicates that about 25% of car crashes have been caused by driver distraction. In the Brazilian scenario, data from the Federal Highway Police show that driver distraction (37%) is the main cause of accidents on federal highways [3]. According to Carney et al. [4], from the reasons that cause driver distraction, attending to passengers (14.6%) and cell phone usage (11.9%) are the top two most common type of distractions.

Smartphones have become more advanced and they include many new features, such as high-speed connectivity, instant messaging, and social networks. This new powerful tool changes the way people interact and socialize, and creates what psychologists are calling smartphone addiction [5]. The last update of the Statistics Portal [6] shows that mobile devices accounted for 48% of web traffic worldwide. This web traffic can be related to the type of application used. Apps such as Facebook, WhatsApp, Instagram, and Messenger are the top four in downloads between Android users and are applications of high social interactivity [7]. In addition to the type of application used, the frequency of use is also a relevant factor to highlight the distraction. According to the survey conducted by Braun Research [8], 70% of drivers say they use the smartphone at some point while driving. Also, in the study conducted by Carney et al. [4], it is revealed that 70% of drivers between 16 and 18 years old have already spoken on the phone while driving, 42% often read a text message or e-mail, and 32% use the cell phone to write a text message while driving.

The degree of danger of mobile phone use while driving depends on the number of simultaneous types of distraction. A driver may be distracted physically, visually, cognitively, or auditively. For example, the simple fact of reading a text message can cause the driver to take their eyes off the road, their hands off the steering wheel, and their mind off the road and the surrounding situation. In this case, the three distractions caused by reading a text message are, respectively, visual, physical, and cognitive distractions.

Texting, reading, and calling while driving are the three types of secondary task that distract the driver due to the use of smartphone. Research by Tran et al. [9], Eraqi et al. [10], Torres et al. [11], Koesdwiady et al. [12] and Streiffer et al. [13] are recent examples of works that propose deep-learning solutions to classify driver distraction. These works employed images from videos to perform macroscopic observation of the driver. The collected image streams are used to classify the driver distraction in function of the observed behavior. Considering the metrics used in these works, all presented good performance to classify the three types of driver distraction concerning the use of the smartphone. However, the use of an external camera can be classified as an intrusive solution and may reduce acceptance in practice. A different non-intrusive approach employs sensor data collected from smartphones. In the survey conducted by Alluhaibi [14], several papers show the use of smartphone sensors to collect data in a non-intrusive way. This approach has a considerable cost-benefit ratio. Today, all new smartphones are equipped with a wide range of sensors such as accelerometer, gyroscope, magnetometer, etc.

According to Liu et al. [15], the bigger challenge is to distinguish between passengers and drivers using only smartphone sensor data. Our claim is that the driver using a smartphone behaves differently than a passenger because the main function must be to drive. Automatically detection of passengers and drivers is necessary to avoid the inconvenience function lock to the passengers, ensuring the effectiveness of the solution. For example, considering a solution that detects texting while driving; if this solution does not distinguish passengers from drivers, two situations can happen:The first situation is to block the typing even if it is a passenger who is texting.The second is to ask whether the user is a passenger or a driver. In that case, if the driver is already using the cell phone while driving, nothing guarantees that he will inform that he is a driver to have his phone locked.

The research carried out by Liu et al. [15], Bo et al. [16], and Wang et al. [17] are examples of detecting texting while driving in a non-intrusive approach. For them, texting while driving is the most dangerous type of distraction because it involves three class of distractions: visual, physical, and cognitive. As much as texting while driving, reading while driving also provides visual, physical, and cognitive distraction, and, as observed by Carney et al. [4] and Schroeder et al. [18], the distraction of reading while driving occurs more often than the distraction of texting while driving. Therefore, we may assume that non-intrusive solutions are needed to prevent the distraction of reading while driving. Considering the presented scenario, this work aims to build a non-intrusive solution that detects reading while driving and classify if a driver or a passenger is reading.

In this work, the task of classifying passengers and drivers in the event of reading while driving is performed from the observation of the effects of the driver behavior. We assume that the driving patterns of the driver change while he is distracted by reading a text message. On the other hand, we understand that if the driver is not distracted, and the passenger is reading a text message, there will be no change in the driver behavior. The observations of the effects of the driver behavior were made from smartphone sensors. We use the accelerometer, gyroscope, GPS, and magnetometer sensors to collect data in a non-intrusive way. We conducted a naturalistic study to collect the samples from the database. The collection process involved 18 volunteers. In total, 11,000 samples were collected, half of the samples represented the readings performed by drivers, and the other half represented the readings performed by passengers (The database is available at https://figshare.com/articles/Passengers_and_Drivers_reading_while_driving/8313620).

The task of distinguishing passengers and drivers in the event of reading while driving is a challenging task. The first reason comes from the short window of time that the event takes place. When a driver holds the phone to read a message, visual, physical, and cognitive distraction should last for a few seconds. We may consider that the task of detecting changes in driver behavior in this short time is a complex problem. The second challenge of this problem concerns the variability of driver behavior observed from the sensors. Being distracted by reading a text message, different reactions may occur. From the more notable reactions such as sudden breaking, to the lighter reactions such as a slight deceleration or change of angular direction. In this sense, to define rules for all possibilities of driver behavior, it would require a significant amount of data representation, and even then, the possibility of error would still be very high.

Considering the nonlinearity of the problem and the need for generalization due to the various possibilities of driver behavior, we understand that the question of reading while driving can be solved by machine-learning models. In this context, the goal of this work is to use only the sensor data from the smartphone with the support of machine-learning models automatically distinguish the passengers and drivers, in the event of reading while driving. In summary, this work evaluates the influence of the sensors Ω and the generalization capacity of the machine-learning models Ψ, in the task of distinguishing passengers and drivers during reading while driving Φ, i.e., we evaluate how Φ can be solved by Ψ(Ω) and also which the influence of ω∈Ω ∀ ψ∈Ψ.

## 2. Data Preparation

The two most popular dataset is the State Farm Distracted Driver Detection [19] and Naturalistic Driving Study (NDS) for the second Strategic Highway Research Program (SHRP 2) [20] projects. However, none of these benchmarks meets the data requirements of our research, which is to have non-intrusive reading while driving data collection using smartphone sensors.

State Farm Distracted Driver Detection was a macroscopic dataset used in a Kaggle competition (https://bit.ly/2MyD4IJ). This dataset was constructed in a naturalistic study where images captured drivers doing something in the car (texting, eating, talking on the phone, makeup, reaching behind, etc.). According to State Farm, the competition aimed to test whether cameras can automatically detect drivers engaging in distracting behaviors. Considered by the literature as intrusive data and little accepted as a practical solution, this dataset is outside the scope of our work because it is macroscopic data that captures images of the driver while driving.

The NDS from the Strategic Highway Research Program (SHRP 2) is the study of naturalistic driving behaviors that monitored approximately 3400 participant drivers between 2010 and 2013. According to Hankey et al. [21], vehicles were instrumented with a data acquisition system (DAS) that collected four video views (driver’s face, driver’s hands, forward roadway, rear roadway), vehicle network information (e.g., speed, brake, accelerator position), and information from additional sensors included with the DAS (e.g., forward radar, accelerometers). Although SHRP2 is a non-intrusive benchmark, it was not built specifically to monitor the reading while driving. Thus, if we chose this benchmark, it would require meticulous data processing to filter only the samples of reading while driving.

In this context, to distinguish passengers and drivers over reading while driving, we have chosen to build our dataset from collecting naturalistic data using smartphone sensors. Our dataset was collected in 2018 in the cities of Belém and Paragominas, cities located in the state of Pará, northern Brazil. Data collection involved 18 volunteers who used our application to capture sensor data while reading text messages. In total, 11000 samples were collected, half of the samples represented the readings performed by drivers, and the other half represented the readings performed by passengers. The Figure 1 illustrates the process of data acquisition and feature extraction performed in this work.

### 2.1. Data Acquisition

To label each message, we have built an Android app to display random short messages to the user. While the user reads a message, data is collected using the accelerometer, GPS, magnetometer, and gyroscope sensors. Figure 2 illustrates the four application screens. In the first screen, top left, the user must press the start button to start the data collection. Then, in the top right screen, the user must define the name of the file that will store the sensor data. To read a message generated by the application, the user must press the begin button; at this point, the sensor data begins to be collected. At the end of the reading, the user must press the end button which informs that the collection of the sensor data can be interrupted and stored.

For each read message, the user must repeat the procedure of pressing the begin/end buttons to store the sensor data. Each message that the user reads represents an instance of collected data. A file has multiple instances of data. The data labeling is performed per file, i.e., when the user creates a file, it is accepted that all read messages will be performed by only one user profile that can be a driver or a passenger. A user can create as many files as he needs.

In the data collection phase, each participant created at least two files per category. In total, 58 files were created, 36 files containing sensor data instances collected while drivers read text messages and 22 files containing sensor data instances collected while passengers read text messages. The participants had the autonomy to decide the appropriate time to read a text message. The only prerequisites established were: (1) To read the text message only when the car is in movement, and (2) To read the text message by holding the smartphone with one or both hands.

The first prerequisite was to discard text messages read when the car was stationary. Furthermore, as the classification of passengers and drivers was based on the sensor data of the vehicle, no differentiation would be observed in this context.

The second prerequisite was to eliminate user interaction when the smartphone was attached to the vehicle. We understand that a fixed smartphone has similar distraction characteristics as an onboard computer and, therefore, data collections in this configuration were not taken into account. As future work, we intend to investigate whether it is possible to distinguish passengers from drivers when they interact with a smartphone attached to the vehicle.

The data collection began in June 2018 and ended in December 2018. During this period, the 18 participants made 58 trips to collect data. All participants collected data assuming the role of driver and passenger. In Table 1, we can observe the number of messages read by each volunteer. Each line represents the collection performed by a volunteer. The passenger column informs the number of messages read by a volunteer with the passenger profile and the driver column the number of messages read by a volunteer with the driver profile. The subdivision of the passenger and driver column informs the number of trips performed by a volunteer. For example, volunteer 1 conducted a trip with the passenger profile and three trips with the driver profile. On the trip as a passenger, volunteer 1 read 103 messages. In three trips as a driver, volunteer 1 read, respectively, 26, 38, and 59 messages. For each reading, the GPS, magnetometer, gyro and accelerometer data were collected.

The number of messages collected on a trip was different among participants. Distance traveled and traffic conditions were two important factors that influenced the number of data collected on each trip. The amount of travel was also different among participants. In this case, because it was a voluntary data collection, a participant had the autonomy to decide how many trips he would like to make. Looking at Table 1, we can observe that most participants opted to perform a larger number of trips as a driver.

In Figure 3, we can see six trips made by participants. Figure 3a–c are trips performed by drivers and Figure 3d–f are trips performed by passengers. Because it was a naturalistic study, the participants could choose the route of the trip. In the chosen route, with the car in movement, whenever possible, the participant read a message to collect sensor data. As we can see in Figure 3, the data were collected on different routes with different directions. This feature is important to not to skew the collection of some sensors.

Another important point that should also be highlighted is how the participants should read the message. We wanted to have as little control as possible in how the data would be collected. For this reason, the participants were instructed to hold smartphones as usual. Some participants held the smartphone with their right hand, others with the left hand, others with both hands. Some held the smartphone horizontally and others vertically. Accepting these differences is important in order to test the generalization capacity of machine-learning models.

### 2.2. Feature Extraction

As previously mentioned, the data collection was performed using the accelerometer, gyroscope, magnetometer, and GPS sensors of a smartphone. The choice of these sensors was due to their use in related work. As noted by Alluhaibi et al. [14], several researchers are investigating driver behavior from data collected by smartphones. The research carried out by Ferreira et al. [22] is an example of related work that used the same set of sensors selected for the data collection of our work. In his research, Ferreira et al. analyzed the influence of the sensor set on the task of classifying driver behavior profiling.

In the context of driver distraction, there are studies that also analyze distraction using data collected by other sensors. For example, the eye tracking sensors, wearable sensor network, galvanic skin response, head tracking system, electrocardiogram (ECG) and surface electromyogram (sEMG) are also used in related work [23,24,25,26]. Using more specific sensors, these studies aim to analyze driver distraction from different perspectives. Cognitive analysis, eye movements, head movements, body reactions, and heartbeats are taken into account to infer the reason and type of driver distractions. Research that seeks this level of detail usually performs experiments in simulators. For example, Pavlidis et al. [27] performed the simulation experiment to study the effects of cognitive, emotional, sensorimotor, and mixed stressors on driver arousal. These studies, although presenting important results, have their capacity limited because they need specialized hardware. For this reason, in order to provide a convenient driver attention solution, we have chosen to use only the sensors of mobile phones.

In our work, with the exception of the GPS sensor, the sensors were configured to collect data at a frequency of 5 Hz. For these sensors, all values corresponding to the x, y, and z-axes were taken into account during data collection. In the case of the GPS sensor, the frequency was 1 Hz, and the data collected correspond to the geographical coordinates of latitude and longitude. All sensor axes were collected since one of the objectives of this work is to verify the influence of these sensors as inputs for machine-learning models. Table 2 lists the 11 data types collected. Due to the frequency setting of the sensors, for each read text message, the data window has different sizes. The size of the data window is directly related to the time that the participant takes to read a message. This time may be associated with the message size and the surrounding distraction reasons. Figure 4 shows the distribution of driver and passenger data windows according to its size. We can see that the variation of the window size of drivers is more significant than the passenger window size. This behavior can be justified due to the level of distraction of the driver. When reading a message while driving, a driver tends to become more distracted than a passenger and, for that reason, the size of the data window tends to be larger.

Figure 4 shows that the average window size of the drivers is larger than the passenger windows. This behavior can be explained due to the focus of the task performed. When a driver is driving and decides to read a text message, reading becomes a secondary task. His visual, physical, and cognitive attention will be shared between the task of driving and reading the text message. On the other hand, when a passenger decides to read a text message, this is probably their primary task. Their concentration is total, and for this reason, tends to perform a faster reading than a driver. Considering the analysis of the data window, we performed the feature extraction from the mean and standard deviation of each sensor 3-axis collected. In the case of latitude and longitude coordinates, we use them to calculate the velocity in *m*/*s* of each data window. Table 3 shows the 19 attributes computed from the four sensors used in the data collection.

Whereas we want to evaluate how Φ can be solved by Ψ(Ω) and also which the influence of ω∈Ω ∀ ψ∈Ψ. In this formulation, the set Ω consists of the attributes of Table 3.

Figure 5 shows a violin plot of the distribution of the data of each attribute. In each plot, the left-hand side represents the distribution of the sensor data collected by the drivers and the right-hand side the distribution of sensor data collected by the passengers. To simplify the visualization, we apply MinMax normalization to all attributes. In this way, all the attributes were rescaled to the range of values between 0 and 1. A violin plot is a hybrid of a box plot and a kernel density plot. The plots of Figure 5a correspond to the attributes collected by the accelerometer and GPS sensors. For these attributes, we can observe that the density of the data of the drivers and passengers are almost symmetrical, in addition, the interquartile are also close. The only exception is the accX-mean attribute data. In this attribute, the driver data density is more dispersed when compared to passenger data. In Figure 5b we can see the plots of the attributes collected by the gyro sensor. In these plots, the attributes computed by the mean have very close density and interquartile. This behavior suggests that the sensor data collected by passengers and drivers are similar. For the attributes computed by the standard deviation, we can see that the driver data density is slightly more dispersed than the passenger data density. By analyzing the plots of the attributes collected by the magnetometer sensor (see Figure 5c), we can see that these are the data that have the greatest difference in density and interquartile. The attributes computed by the mean have dispersed and asymmetrical density and different interquartile between the data of the passengers and the drivers.

Analyzing the distribution of the data, we can infer that the attributes corresponding to the magnetometer are the best to try to distinguish passengers and drivers. The magnetometer measures the force of the magnetic field applied to the device in micro-Tesla (μT), and works like a magnet.

Despite the asymmetry in the distribution of the magnetometer data, we can see that the difference in data from drivers and passengers is not too significant. This behavior suggests that the classification of drivers and passengers using only sensor data is not a trivial task. Statistical analysis or threshold algorithms would probably have difficulty in this task. For this reason, in this paper we intend to evaluate the performance of different models of machine learning in the task of distinguishing passengers and drivers while reading text messages.

## 3. Machine-Learning Models

The Φ objective, which is the task of distinguishing passengers and drivers in the event of reading while driving, can be seen as a binary classification problem. Due to the characteristics of nonlinearity of data, it is a problem that can be solved with machine-learning models. We built seven machine-learning models to compose the set Ψ. We take into account distinct supervised learning strategies, namely three eager-learner (SVM: Support Vector Machines, DT: Decision Tree and LR: Logistic Regression), three ensemble learners (RF: Random Forest, ADM: AdaBoost Machine and GBM: Gradient Boosting Machines), and one deep learning (CNN: Convolutional Neural Network). We chose the learning models from the literature review. In this review, we selected primary papers that performed some experiment to analyze or prevent the driver distractions due to the use of mobile phones. In the conducted literature review, we use repositories ACM Digital Library, ScienceDirect, and IEEE Xplore Digital Library. The conduction of the systematic review was performed to answer the following research questions:Question 1: What types of methods are used for the analysis or detection of driver distraction?Question 2: What are the types of studies performed for the analysis or detection of driver distraction?Question 3: If so, what are the sensors used for data collection?Question 4: What are the techniques used for the analysis or detection of driver distraction?

Eighty-six papers were selected, of which 42 developed machine-learning models to detect or classify driver distraction. Table 4, shows the models observed in the literature review, as well as the number of papers that used them. Eager learning is a strategy in which the system tries to construct an input-independent target function while training the model. SVM, Decision Tree, and Logistic Regression are models that use this learning strategy.

### 3.1. SVM

Support Vector Machines use a linear model to implement nonlinear class boundaries. The basic idea is to create support vectors (line or hyperplanes) to separate the target classes. To solve a nonlinear problem, the model applies several transformations in the data via a mapping function α and train a linear SVM model to classify the data in this new to a higher-dimensional feature space. According to Raschka and Mirjalili [56], one problem with α mapping approach is that the construction of the new features is computationally costly, especially if we are dealing with high-dimensional data. To avoid these expense, SVM solutions use a kernel function. In the construction of our model, we use the Radial Basis Function (RBF) kernel function (see Equation (Equation 1)).
(1)$$\kappa \left(x^{\left(i\right)},x^{\left(j\right)}\right)=exp\left(-\frac{∥x^{\left(i\right)}-x^{\left(j\right)}{∥}^{2}}{2\sigma^{2}}\right)$$

### 3.2. Decision Tree

A decision tree is composed of several layers of *testing nodes* (testing some attribute), *branches* that represents the outcome of the testing nodes and the *leaf nodes*. The leaf nodes represents the class label in classification problems. The path from root to leaf represent rules. Tree-based predictive models have high accuracy, stability and ease of interpretation.

Decision tree classifiers are attractive models if we care about interpretability. In a tree, the data split (testing node) is performed by information gain (IG). The goal of IG is to split the nodes at the most informative features to the samples at each node all belong to the same class. Equation (Equation 2), shows the computation of the IG of a binary tree. Here, *f* is the feature to perform the split, *I* is the impurity measure, $D_{p}$, $D_{left}$, and $D_{right}$ are the data set of the parent node, the left child node, and the right child node, $N_{p}$ is the total number of samples at the parent node, $N_{left}$ and $N_{right}$ are the numbers of samples in the left and right child’s nodes.
(2)$$IG\left(D_{p},f\right)=I\left(D_{p}\right)-\frac{N_{left}}{N_{p}}I\left(D_{left}\right)-\frac{N_{right}}{N_{p}}I\left(D_{right}\right)$$

### 3.3. Logistic Regression

Logistic regression is a discriminative probabilistic model. It models the posterior probability distribution P(Y|X), where *Y* is the target variable, and *X* is the features. Given *X*, they return a probability distribution over *Y*. Figure 6 shows a typical architecture of the logistic regression model. In a binary classification problem, the output of the sigmoid function is interpreted as the probability of a particular sample belonging to positive class, i.e., ϕ(z)=P(y=1|x;w) where *z* is the linear combination of weights and samples features $z=w^{T}x$.

Logistic regression is one of the most widely used algorithms for classification. In our literature review, we observed that logistic regression is mainly used in studies whose main area is not computation. Works by Yannis et al. [42], He et al. [43] and Haque et al. [41] applied logistic regression to investigate the driver distraction due to the use of the cell phones. In our study, we consider logistic regression as a baseline model. After the construction and evaluation of other machine-learning models, if no model demonstrates better performance, then, by the simplicity of the logistic regression model, this will be the one chosen to distinguish drivers and passengers in the event of reading while driving.

The second strategy of model construction addressed in this work was the ensemble-learning. The main idea of this strategy is to combine different classifiers into a meta-classifier that has better generalization performance than each classifier alone. The ensemble-learning built in this work were Random Forest, AdaBoost, and Gradient Boosting.

### 3.4. Random Forest

According to Breiman [57], a random forest is a collection of tree-structured classifiers $\left\{h\left(x,\Theta_{k}\right),k=1,...\right\}$ where the $\left\{\Theta_{k}\right\}$ are independent distributed random vectors and each tree casts a unit vote for the most popular class at input *x*. In this context, a random forest can be considered to be a bagging of decision trees. The idea behind a random forest is to have multiple decision trees to build a more robust model that has a better generalization performance, better stability and lower susceptibility to overfitting. Instead of searching for the essential feature while splitting a node, the random forest algorithm searches for the best feature among a random subset of features. This results in a wide diversity that generally results in a better model. Another high quality of the random forest is that it is straightforward to measure the relative importance of each feature on the prediction. In the evaluation phase, where employ RF to verify the influence of ω∈Ω ∀ ψ∈Ψ, i.e., to measure the feature importance and reevaluate the behavior of the models with the subset of more essential features.

### 3.5. AdaBoost

The central idea of Adaptive Boosting is to create a strong classifier from some weak classifiers. The main difference from the Random Forest approach is that AdaBoost creates several random weak models that do not interact during creation of the trees, they run completely separable. The trees are used to vote and the majority wins. AdaBoost proceeds in rounds. For each round, the algorithm choose training sets for the base learner in such a fashion as to force it to infer something new about the data each time it is called. During the rounds, AdaBoost maintains a distribution over the training examples. The distribution used on the *t*-th round is denoted $D_{t}$, and the weight it assigns to training example *i* is denoted $D_{t}\left(i\right)$. As observed by Schapire and Freund [58], this weight is a measure of the importance of correctly classifying example *i* on the current round.

During the training, hard examples get a higher weight. Higher $D_{t}\left(i\right)$ to hard cases enable the learner to focus its attention on them. The final classification equation used by AdaBoost is represented by Equation (Equation 3) which is the weighted combination of *M* weak classifiers. Here, $f_{m}$ stands for the $m_{th}$ weak classifier and $\theta_{m}$ is the corresponding weight.
(3)$$F\left(x\right)=sign\left(\sum_{m=1}^{M}\theta_{m}f_{m}\left(x\right)\right)$$

### 3.6. Gradient Boosting

As well as adaptive boosting, gradient boosting combines weak learners into a single strong learner in an iterative fashion. The objective is to minimize the loss of the model by adding weak learners using a gradient descent-like procedure. The loss function is a measure indicating how good model coefficients are at fitting the underlying data. Similar to other boosting algorithms, Gradient Boosting builds the additive model in a greedy fashion (see Equation (Equation 4)).
(4)$$F_{m}\left(x\right)=F_{m-1}\left(x\right)+h_{m}\left(x\right)$$

In this equation, $F_{m-1}$ is the previous ensemble and $h_{m}$ represents the newly added base function for tries to minimize the loss *L*. Given the loss function *L*, $h_{m}$ is computed according to Equation (Equation 5) where $y_{i}$ is the target label.
(5)$$h_{m}=arg\min_{h}\sum_{i=1}^{n}L\left(y_{i},F_{m-1}\left(x_{i}\right)+h\left(x_{i}\right)\right)$$

### 3.7. Convolutional Neural Network

The Convolutional Neural Network (CNN) is a type of deep learning that became popular due to its excellent performance in visual objects recognition and classification problems. In the literature, there are several proposals of convolutional neural network architectures. Architectures such as LeNet-5, AlexNet, ZFNet, VGGNet, GoogleNet, and ResNet show the evolution of convolutional neural networks and the trend of even deeper architectures. Regardless of the architecture and depth, it is possible to observe that there is a pattern of the components used in the construction of a CNN. According to Goodfellow et al. [59], a convolutional layer is formed by 3-tuple *<convolution, activation function, pooling>*.

Convolution is a mathematical theorem applied on two functions *f* and *g* to obtain a third function *h*, defined according to Equation (Equation 6).
(6)$$\left(f*g\right)\left(c\right)=h\left(c\right)=\sum_{a}f\left(a\right)\cdot g\left(c-a\right)$$

On a CNN, convolution works as feature maps. Each convolution has the function of learning specific characteristics of the input data. This learning takes place through the adjustments of the weights, during the training of the network. The result of this local weighted sum is then passed through an activation function. The convolution of a CNN is a linear system. In this sense, multiple convolutions also form a linear system. In order for a CNN to solve non-linearly separable problems, it is necessary to use nonlinear activation functions among each convolution layer. In the context of the convolutional neural network, studies such as Gu et al. [60], Alcantara [61] and LeCun et al. [62] point out that the Rectified Linear Unit (ReLU) function and its variations have good performance for the neurons of the convolution layers. The ReLU function (see Equation (Equation 7)) calculates the maximum value of the input values and transforms all the negative input values to zero.
(7)f(x)=max(0,x)

The last component of 3-tuple is the pooling function. These functions analyze a set of neighbors and extract a characteristic that represents them. The goal is to merge semantically similar features and make feature mapping invariant, even if there are variations in input values. The max pooling function proposed by Zhou and Chellappa [63] performs this representativity by extracting the maximum value among the observed neighbors. In addition to max pooling, Gu et al. [60] emphasizes that the functions lp pooling, mixed pooling, stochastic pooling, spectral pooling, spatial pyramid pooling, and multi-scale ordering are also functions used in CNNs.

Recent studies demonstrate that deep-learning models perform well with sensor and temporal data. The work by Zhang et al. [64] demonstrates that deep neural networks have been proved to be useful in learning from speech data. Sensor data of a smartphone are also kinds of time series with similar characteristics to speech signals. Inspired by this, in this paper, we propose a novel deep-learning approach for distinguishing passengers from drivers during reading while driving.

## 4. Setting up the Hyperparameters

We have two types of parameter’ settings in ML: those that are learned during the training phase of the model and those that need to be preconfigured. The parameters that are preconfigured are called hyperparameters. The configuration of the hyperparameters can be determinant for the performance of the developed model. Hyperparameters are generally configured in three ways: empirically from the experience of the designer; from heuristics; or by brute force.

In this work, we perform the hyperparameter calibration using a grid search. A grid search is an exhaustive search where we specify a list of values for different hyperparameters, and iteratively we evaluate the performance of the model for each combination of those to obtain the optimal combination of values from that set.

Table 5 shows the configuration parameters, and the range of values that we pre-determine for each model. We performed the grid search of the hyperparameters using one thousand stratified samples from our database. The values in bold represent the best configuration of the models after the grid search processing. Logistic regression is our baseline model, we did not submit to the grid search process. In this case, for logistic regression, we use the default settings of the scikit-learn library.

In the SVM model, we set the C, Kernel, and Gamma parameters. The C parameter tells the SVM optimization how much you want to avoid misclassifying each training example. A large value of parameter C makes the model to use lower margin hyperplane if this hyperplane performs a better classification on the training data. A small value of parameter C makes the model to look for a larger margin separating hyperplane making the decision surface smooth. The kernel function determines how the hyperplanes will be constructed. If a linear function is used, then the constructed model will treat the input data as linearly separable. For nonlinear problems, the Gaussian RBF is widely used when there is no prior knowledge about the data. Regarding the Gamma parameter, it defines how far the influence of a single training example reaches. If the value of Gamma is small, the influence of a single example of training in the construction of the hyperplanes is smaller. In the grid search performed, we can observe that the built SVM model uses lower margin hyperplane, it considered the input data to be non-linearly separable and that the influence of a single training sample should not be significant for the construction of the hyperplanes.

In the Decision Tree and Random Forest models, the same parameters were configured. The only exception was the N estimators parameter which is a particularity of the Random Forest model and determines the number of trees in the forest. In the Criterion parameter is set the function to measure the quality of the splits that are realized during the construction of the trees. Max depth, Max features, and Min sample split are parameters used to determine the tree configuration. In the grid search performed, we can see that Decision Tree and Random Forest have been configured to build shallow trees. In relation to the minimum number of samples required to split an internal node, the Random Forest used a large number of samples to perform this task. In the Random Forest model that we build, 80 samples are needed to divide a node. In the Max depth parameter, which concerns the number of features to consider when looking for the best split, the two models were configured with $\sqrt{n}$ where *n* is the number of the training sample.

In the Adaptive Boosting and Gradient Boosting models, we performed the N estimators and learning rate parameters. In the N estimators, we defined the number of boosting stages to perform. The boosting models are fairly robust to overfitting so a large number usually results in better performance. In the learning rate, the set value has the purpose to define the contribution of each classifier. The nearer to zero is the learning rate, it means that the classifiers, individually, will have less contribution to the ensemble, i.e., for a small learning rate, the ensemble will take into account the joint performance of the classifiers. For the Gradient Boost model, the parameters Min sample split and Max depth were also configured. In this case, these parameters have the same purpose as the configuration performed in the Random Forest and Decision Tree models.

In the case of CNN, because it is a neural network, we had to perform the configuration in a hybrid way. The number of convolutional layers and the number of filters in each layer was defined empirically. The size of each filter was specified using the grid search process. In grid search, to find the best configuration of the convolutional filters, we determine the range of values of [1–5]. For each layer, all filters have the same size. As we defined four convolutional layers, 625 combinations were tested to determine the architecture as shown in Figure 7.

The construction of this architecture took into account the observations made by Szegedy and collaborators [65]. In the development of the CNN, the authors demonstrate that the use of small kernels is more efficient than the use of larger kernels. In addition to decreasing the processing load, Szegedy and collaborators [65] also emphasize that the use of multiple small filters can match the representativeness of larger filters. The concept of representativity is related to the capacity of the convolution to be able to detect structural changes in the analyzed object. In this context, we decided to use four convolutional layers with, respectively, 60, 40, 40, and 60 filters in each layer. As already explained, the filter size of each layer was defined from the grid search process.

The construction, configuration, training, testing, and validation of all models was performed using the Scikit-learn API and its dependencies. In the case of the neural network, we also use the Keras API and its dependencies.

## 5. Results

To analyze how ω∈Ω sensors can influence the performance of the ψ∈Ψ models, we divide the experiments into three phases. In the first phase, we used all the features of the Ω set to train, evaluate and test the performance of the seven machine-learning models that belong to the Ψ set. In the second phase, we evaluated the performance of the ψ∈Ψ models using only the features of each sensor. The objective is to verify if any model presents good classification performance using the features of a single sensor. In the last phase of the experiment, we evaluated the performance of the ψ∈Ψ models from the most significant features highlighted by the feature importance analysis of the gradient boosting model. In this third phase, the goal is to verify if it is possible to use a subset of the Ω set to obtain a model ψ∈Ψ that is statistically equivalent to the best model obtained in the experiment of the first phase. If this is possible, it means that we can reduce the amount of information needed to distinguish passengers and drivers in the event of reading while driving.

In all the experimental phases, we have set aside two third of the dataset to performing the training and validation of the model using 10-fold cross-validation methodology. Additionally, to ensure the stability of the models, in each phase, we repeated ten times the process of training and validation of the models using the 10-fold cross-validation methodology. To evaluate the performance of the models, we use the accuracy, precision, recall, F1-score, and Cohen kappa score that are well-known metrics for machine-learning designers.

The accuracy metric checks the proportion of correctly classified samples. When the accuracy reaches its maximum value (=1), it means that the model did not produce false positives and false negatives. The accuracy evaluates the overall performance of the classifier. In that sense, when we have unbalanced classes or a multiclass problem, the good accuracy rates may not reflect the actual performance of the model. For a better evaluation of the model, it is evident that besides the global vision of the metric accuracy, there is also a need for metrics such as precision and recall to evaluate the specificities of each class. The precision metric, for a $C_{x}$ class, checks the proportion of samples correctly classified as $C_{x}$ based on the examples that have been classified as $C_{x}$. In a classification problem, good precision rates indicate the correctness of the classifier. About the recall metric, this checks the proportion of samples that were classified as $C_{x}$ in function on the samples that should be classified as $C_{x}$. A good recall rate (≈1) indicates the completeness of the classifier.

In the classification task of detect passengers and drivers in the event of reading while driving, a good precision rate in the classification of drivers indicates the non-inconvenience of the passengers, i.e., how much better the precision of the model, the smaller the number of passengers who will have their cell phone blocked because it has been wrongly classified as a driver. For the recall metric, your good rates indicate the security of the system. In the ideal scenario, recall=1 means that all drivers have been correctly classified. To check the balance between non-inconvenience and system security, we used the F1-score metric. The F1-score is the harmonic mean of precision and recall.

The last metric used was the Cohen kappa, which is a robust statistic useful for reliability tests between raters. It can range from −1 to +1, where values ≤0 represent the amount of agreement that can be expected from the random chance, and 1 represents the perfect agreement among the evaluators. In the machine-learning perspective, we can use kappa to measure how closely the instances labeled by the classifiers matched the data labeled as ground truth.

### 5.1. Evaluations of Machine-Learning Models with All Features

Based on the methodology described in the Section Machine-Learning Models, Figure 8 and Figure 9 show the performance of the models in the first phase of the experiment. We built the boxplot using the validation performance of the training phase. The boxplot was constructed with one hundred validation performances obtained with the ten replications of the 10-fold cross-validation. We can observe that the Gradient Boosting and CNN models obtained the best performance in all analyzed metrics. The Decision Tree model presented similar behavior to our baseline model (Logistic Regression). Also, for almost all metrics, the Decision Tree (DT) presented the most significant tail. This behavior tells us that DT was the model that presented greater instability in the classification of drivers and passengers. The ensemble models Random Forest and AdaBoost, and eager learning SVM performed well, but they were not as effective compared to the Gradient Boosting and CNN models. In the specific case of the SVM model, we can observe that it presented little disparity in the recall metric and a considerable difference in the precision metric. This behavior shows us that the SVM model was better to ensure the safety of drivers than to avoid the inconvenience of passengers.

Figure 9 shows the radar chart of the test phase (left) and the validation phase (right). As we can see, for all metrics, all models have kept their performance in the test and validation phase. This behavior demonstrates that no model has specialized only in the training set, and each one with its level of performance, was able to realize the generalization of the problem. About the generalization capacity, we can classify the models into three levels. At the first level are the Gradient Boosting and CNN models as the models with the highest generalization capacity. At the second level are the AdaBoost, Random Forest, and SVM models as models with intermediate generalization capability. Finally, in the third level, we have the Decision Tree and Logistic Regression models as the models with the least generalization capacity of the problem.

### 5.2. Evaluations Using the Features of a Single Sensor

In the second experiment, we analyzed the performance of the models taking into account only the parameters collected from the same sensor. We divide the set of sensors Ω into three subsets. The first subset contains the parameters collected from the accelerometer sensor. The second subset includes the parameters obtained from the gyro sensor. The third subset contains the parameters collected from the magnetometer sensor. Table 6 shows the parameters belonging to each subset. At this stage of the experiment, we did not use the speed feature calculated from the data collected by the GPS sensor because it was the only information coming from the GPS. In this case, we have analyzed that single speed is not enough information to distinguish passengers from motorists in the event of reading while driving.

Figure 10, Figure 11 and Figure 12 show the performance of the models for each subset of parameters specified in Table 6. For each subset, we plot the validation data boxplot and the radar chart of the validation and test data. In this phase, the training of the models was also carried out by running ten times the 10-fold cross-over estimation method.

Figure 10 plots the performance of the models with the features collected by the accelerometer sensor (subset 1). Figure 11 shows the performance of the models with the features collected by the gyroscope sensor (subset 2). Figure 12 shows the performance of the models when analyzed with the data collected by the magnetometer sensor (subset 3).

Analyzing the boxplots from Figure 10 and Figure 11, we can see that the performance of the models were lower when compared to the experiment that used all the features of the Ω set. In the context of driver behavior, the combination of the accelerometer and gyroscope sensors allows us to identify changes in acceleration, curves, and braking patterns. However, when analyzing the data from these sensors separately, we can note that the independent use of the accelerometer and the gyroscope sensors are not enough to distinguish passengers and drivers in the event of reading while driving. McHugh [66] suggests that kappa values less than 0.81 are outside the perfect agreement range of this metric. Thus, we can observe that in the analysis of subsets 1 and 2, all models are with kappa values below the expected one.

Figure 12 shows the performance of the models for the magnetometer sensor data. We can see in the boxplot that the performance of the Gradient Boosting and CNN models were similar to those verified in the first experiment. Regarding the Decision Tree model, although it is the model that presents the most substantial tail in all metrics, we can notice a considerable improvement in its performance when compared to its performance of the first experiment. On the other hand, the SVM model that was in the second stage of performance in the first experiment, this time presented results close to that of our baseline model. In the radar chart of the validation and test, we can see that the models Gradient Boosting and CNN presented the pentagon that indicates the capacity of generalization of the model.

In this phase of the experiment, observing the similarity of performance of the GB and CNN in both the first and second phases of the research, we decided to apply a statistical significance test to verify the equivalence between the models. We call $GB_{1}$ and $CCN_{1}$ the Gradient Boosting and Convolutional Neural Network models trained with all the features of the Ω set and of $GB_{2}$ and $CNN_{2}$ the models trained only with the subset of features derived from the magnetometer sensor. First, we generate the kernel density estimation of the accuracy, and we apply the normal test to make sure that all models have a normal distribution with 95% confidence. Assuming that all distributions are normal, we apply the Student’s *t*-test. We want to verify if the developed models are significantly equaled, the null hypothesis of the test (H0) considers that the models are equal and the alternative hypothesis (H1) considers that the models are different. If we accept the null hypothesis, it means that there is no significant difference between the compared models. When applying the Student’s test, if we get a *p*-value ≤0.05, it means that we can reject the null hypothesis and the machine-learning models are significantly different with 95% confidence. We apply the test for all two-to-two combinations of the models. As we are comparing four models, then we have performed six hypothesis tests. Table 7 shows the *p*-value for each test and the description whether the hypothesis H0 was accepted or rejected.

We can see in Table 7 that only the comparison of the Gradient Boosting models had their hypothesis rejected. The rejection of the H0 hypothesis in the comparison between $GB_{1}$ and $GB_{2}$ showed us that Student’s test is not distributive. For example, the fact that $GB_{1}$ is equivalent to $CNN_{1}$ and $CNN_{1}$ is equivalent to $GB_{2}$ did not imply equivalence of $GB_{1}$ and $GB_{2}$. If we take into account the performance of the models, the amount of information required, and statistical equivalence. We can say that the model $CNN_{2}$ is the most suitable to carry out the classification of drivers and passengers in the event of reading while driving. $CNN_{2}$ is the best model since it is statistically equivalent to all others, presents a similar performance for all analyzed metrics, and only requires information from the magnetometer sensor to perform the classification.

### 5.3. Seeking for the Best Set of Attributes

In the third phase of the experiment, we used the random forest model to calculate the feature importance score of the Ω set. In a random forest, the idea is that the calculation used to determine the node of a decision tree can be used to evaluate the relative importance of this node about the predictability of the target variable. In this sense, features used at the top of the tree are more important to the final prediction decision. In scikit-learn, the library used in this research, the mean decrease in impurity (MDI) is used to estimate the feature importance in a random forest. According to Louppe [67], in the context of ensembles of randomized trees, the importance of a feature $X_{j}$ can be evaluate by adding up the weighted impurity decreases $p\left(t\right)\Delta_{i}\left(s_{t},t\right)$ for all nodes *t* where $X_{j}$ is used and getting averaged over all *M* trees in the forest. The Equation (Equation 8) calculates the MDI of a feature. In this equation, $X_{j}$ represents the feature that we want to calculate the importance; *M* represents the number of trees in the forest; *t* is a node of the tree; p(t) is the proportion of samples reaching *t*; and $\Delta_{i}\left(s_{t},t\right)$ is the impurity of *t* which is calculated by some impurity measure, such as e Gini index and Shannon entropy.
(8)$$Imp\left(X_{j}\right)=\frac{1}{M}\sum_{m=1}^{M}\sum_{t\in m}p\left(t\right)\Delta_{i}\left(s_{t},t\right)$$

We can see the importance of each feature of the Ω set in the Figure 13. The six most important features are: {1: gyroz_std, 2: magx_mean, 3: magx_std, 4: gyroz_mean, 5: magz_mean, 6: magy_mean}. Considering that in the second experiment we were able to construct a model that uses six features and presents an equivalent performance to the model that uses all the features of the Ω set, in this third experiment our objective is to verify if it is possible to construct an equivalent model using six characteristics or less. Thus, we train and compare the performance of the models taking into account the six most important features. Then, we repeat this procedure for the five most important features and then for the four most important ones. We did not evaluate the top 3 subset because we observed the performance drop in the top 6, top 5 and top 4 experiments.

In Figure 14 we can observe the radar chart of the models. From a global perspective, we can note that model performance decreases as the set of features also decreases. In the radar chart of the top 6 and top 5 subsets, we can observe the performance hierarchy that also is seen in the previous experiments. However, in the analysis of the top 4 subset, the performance hierarchy is non-existent. This behavior and the performance drop across all metrics show that the models cannot generalize the problem using only the top 4 subset as input data. Analyzing the boxplot of the top 6 subset, we can observe that the quartiles of the two best models are closer than in the previous experiments. This means that the Gradient Boosting and Convolutional Neural Network models present lower performance variation using the features of the top 6 subset as input data. In the radar chart, we can see that the GB and CNN models show similar behavior in all five metrics analyzed. To certify this similarity, we applied the statistical test to verify the equivalence between the models. In addition to this comparison, we also performed a statistical correlation with the best models of the previous experiments.

Table 8 shows the results of the statistical equivalence test applied to the models. We use the kernel density estimation of the accuracy in comparing the models. After certifying that all distributions are normal, we apply Student’s *t*-test. The null hypothesis of the test (H0) considers that the models are equal and the alternative hypothesis (H1) considers that the models are different. As in the second phase experiment, here, we also use 95% confidence, i.e., if we get a *p*-value ≤0.05, it means that we can reject the null hypothesis and the machine-learning models are significantly different with 95% confidence. In Table 8, we call the two best models of the top 6 experiment of $GB_{top6}$ and $CNN_{top6}$. The names of the best models of the first and second experiments were maintained as $GB_{1}$, $CNN_{1}$, $GB_{2}$ and $CNN_{2}$.

Although the $CNN_{top6}$ and $GB_{top6}$ show similar performance for all analyzed metrics, the test indicates that statistically, these models are different. However, when comparing the performance of these models with the best models obtained in the first two phases of the experiments, we can see that both are statistically equivalent in almost all comparisons. The only exception was the comparison between the $CNN_{top6}$ and $GB_{2}$ models. The $GB_{top6}$, which is a Gradient Boosting model, was considered equivalent to all the best models of the first two experiments. Thus, taking into account the analysis of the three experiments, we can verify the following evidence:The problem of classifying passengers and drivers in the event of reading while driving can be solved in a non-intrusive way from a machine-learning approach. This fact was demonstrated from the performance of the developed models.Considering all features of the Ω set, the use of the CNN or GB model is statistically equivalent in the task of classifying passengers and motorists in the event of reading while driving.The Ω subset containing only features collected by the accelerometer sensor was not sufficient to construct efficient models for the classification of passengers and drivers in the event of reading while driving.The Ω subset containing only features collected by the gyroscope sensor was not sufficient to construct efficient models for the classification of passengers and drivers in the event of reading while driving.The Ω subset containing only features collected by the magnetometer provided the construction of efficient machine-learning models to classify drivers and passengers in the event of reading while driving. According to the analysis and statistical test, the Convolutional Neural Network was the best model to work with this subset.The Ω subset containing the six main features derived from the Random Forest model provided the development of efficient models to classify passengers and motorists in the event of reading while driving. According to the analysis and statistical test, Gradient Boosting was the best model to work with this subset.To K≤5, the Ω subset containing the *K* main features derived from the Random Forest model was not enough to build useful models for the classification of passengers and drivers in the event of reading while driving.

## 6. Conclusions

Studies show that driver distraction is an increasing problem for road traffic injury and deaths. In this context, this research presented a machine-learning analysis to solve the problem of reading while driving in a non-intrusive way. To analyze the effects of distraction, we collect the information using the GPS, accelerometer, magnetometer, and gyroscope sensors of the smartphone. We have built an app to perform the dataset labeling. The application generated random messages that should be read by the user. During reading, we were able to store the sensor values and label if a driver or passenger read the message. In total, we built a dataset with 11,000 samples. We split the database into three disjoint sets: training, validation, and testing. The training and validation sets were divided and applied in the 10-folds cross-validation process. After each phase of the experiment, we used the test set to verify the generalization capability of the models. Three experiments were carried out. The objective was to verify the influence of the sensors Ω and the generalization capacity of the machine-learning models Ψ, in the task of distinguishing passengers and drivers during reading while driving Φ, i.e., we evaluate how Φ can be solved by Ψ(Ω) and also the influence of ω∈Ω ∀ ψ∈Ψ. In the first experiment, we verified the generalization ability of the models taking into account all the features of the set Ω. The results of this experiment showed that CNN and GB models had the best performance. In the second experiment, we analyzed the performance of the models considering as input only the features collected by the same sensor. Thus, we looked at how the models behaved taking as input data features from the accelerometer, magnetometer, and gyroscope sensor. The results show that the models did not generalize the problem only with the accelerometer or gyroscope information. Regarding the magnetometer data, the results show that CNN had the best generalization. The statistical tests also pointed out the equivalence between this CNN model and the two best models of the first experiment. In the third experiment, we used the Random Forest model to verify the importance of the features. Considering the most important features, the idea was to find a subset of Ω that could be used to construct equivalent or more efficient models than the first experiment. Since we had already found an equivalent model in the second experiment and this model used only six features as input data, in this third experiment we were looking to find a subset of features ≤6. The results showed that the subsets ≤5 were not sufficient to construct a model with similar performance to the best models of the first experiment. However, with the subset of the six most essential features obtained by the Random Forest model, it was possible to construct a Gradient Boosting model statistically equivalent to the best models purchased in the previous experiments. Taking into account the results presented in this research, we can conclude that CNN and GB are efficient models to distinguish passengers and drivers in the event of reading while driving. The fact that we build models that work with non-intrusive data indicates the possibility of making commercial solutions that can provide driver safety and convenience for passengers. As future work, we intend to analyze the effects of reading while driving when the device is attached to the vehicle. With this approach, we can apply the analysis to other devices such as the onboard computer.

## Figures and Tables

**Figure 1 sensors-19-03174-f001:**
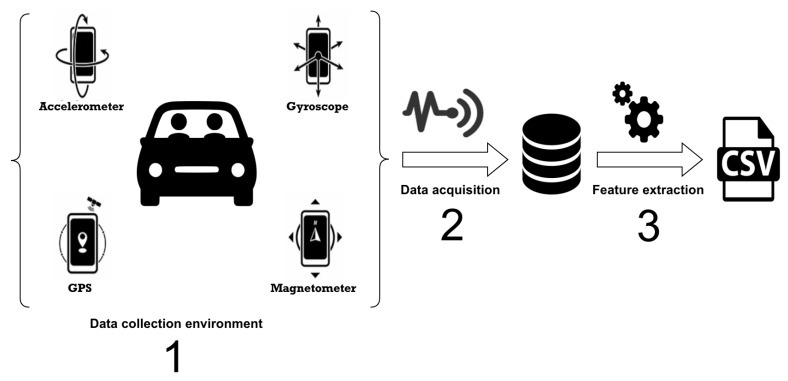
Abstract view of dataset acquisition and feature extraction.

**Figure 2 sensors-19-03174-f002:**
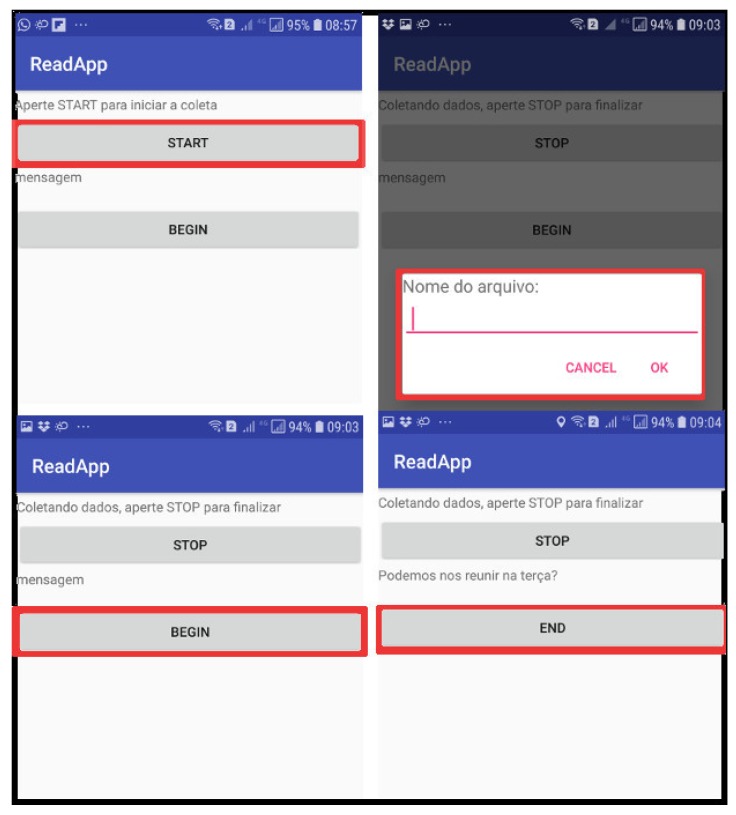
Data Acquisition App: In the first screen (**top right**), the user must enter the name of the log file. The sensors are activated when the user presses the START button. For each message read, the user must press the begin/end buttons to set the sensor data window.

**Figure 3 sensors-19-03174-f003:**
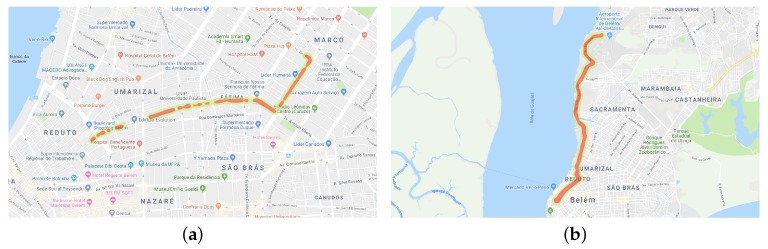
Trips made by volunteers to collect sensor data. Figures (**a**–**c**) were trips performed by drivers and Figures (**d**–**f**) were trips conducted by passengers.

**Figure 4 sensors-19-03174-f004:**
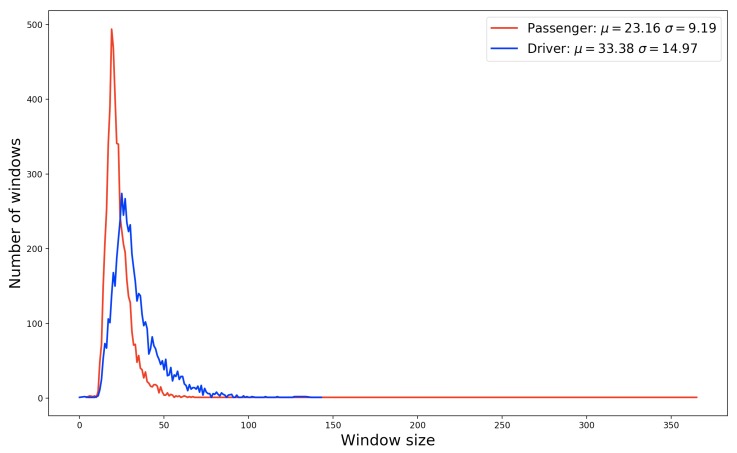
Distribution of windows size collected by users. Each data window represents the interval that the user took to press the begin/end buttons of the app.

**Figure 5 sensors-19-03174-f005:**
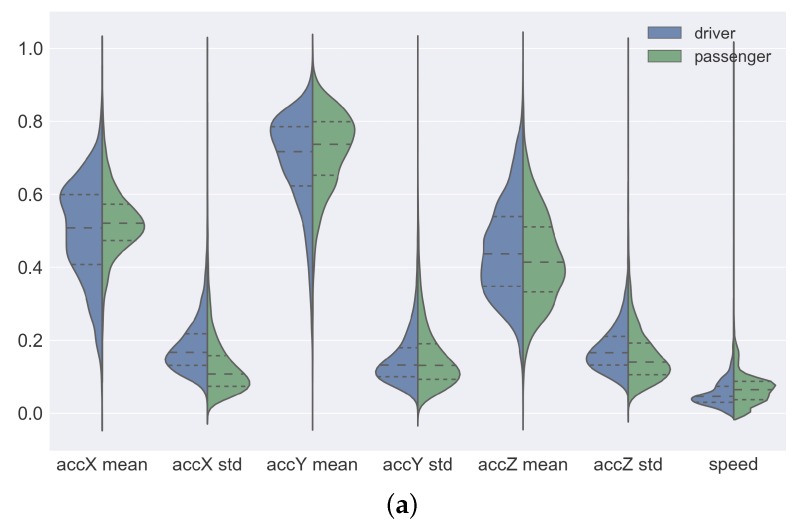
Violin Plot of the distribution of features computed from the sensor data. (**a**) Violin Plot of the distribution of features of accelerometer and GPS sensor data. (**b**) Violin Plot of the distribution of features of gyroscope sensor data. (**c**) Violin Plot of the distribution of features of magnetometer sensor data.

**Figure 6 sensors-19-03174-f006:**
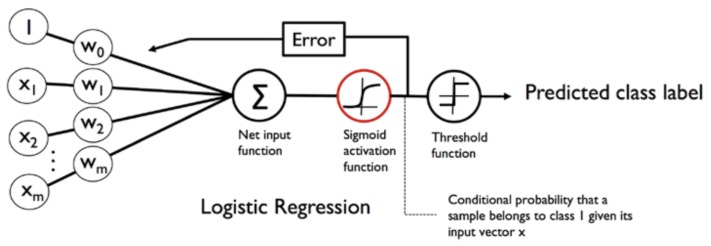
Architecture of a Logistic Regression Model [56].

**Figure 7 sensors-19-03174-f007:**
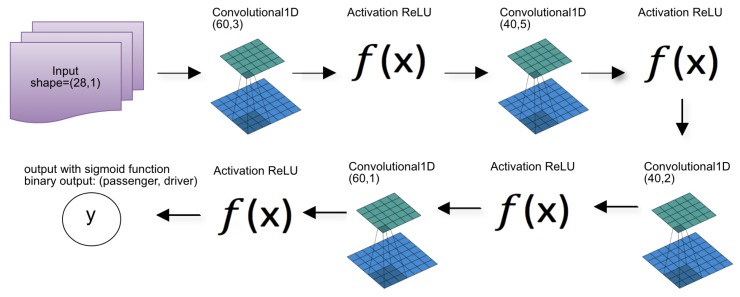
CNN architecture developed in this work.

**Figure 8 sensors-19-03174-f008:**
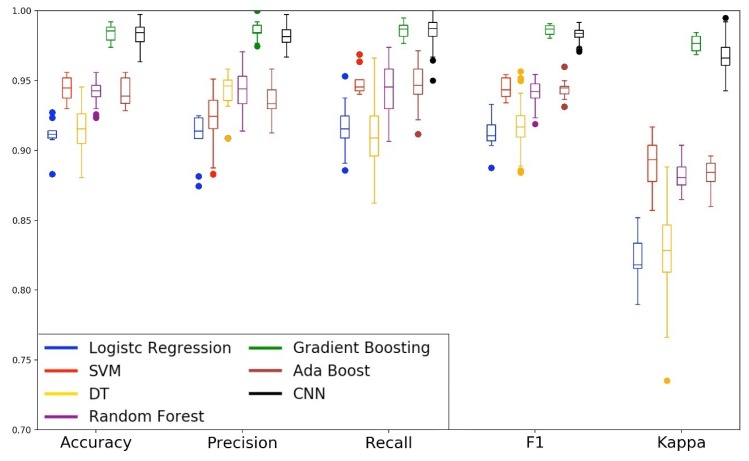
Box plot of the first experiment.

**Figure 9 sensors-19-03174-f009:**
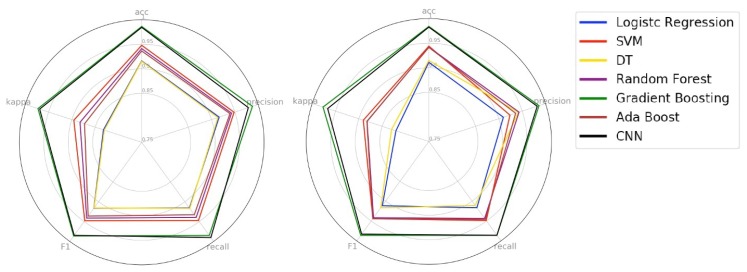
Radar chart of the test phase (**left**) and the validation phase (**right**).

**Figure 10 sensors-19-03174-f010:**
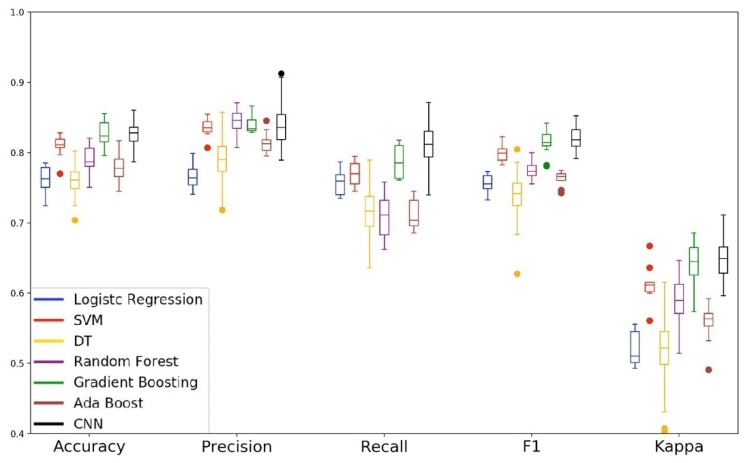
Performance of the models with the features collected by the accelerometer.

**Figure 11 sensors-19-03174-f011:**
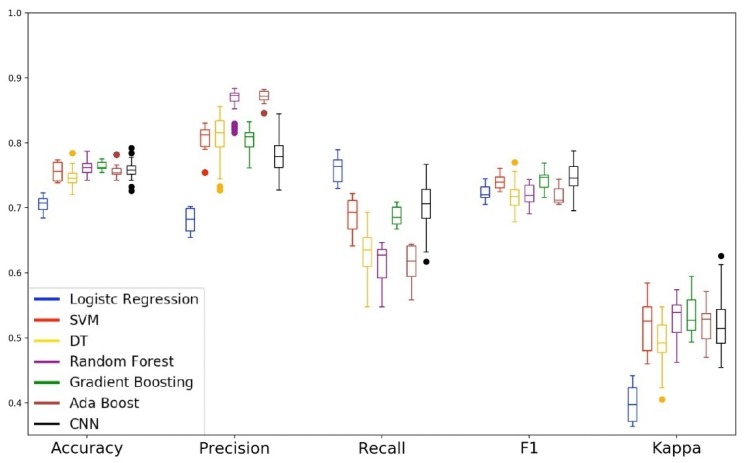
Performance of the models with the features collected by the gyroscope.

**Figure 12 sensors-19-03174-f012:**
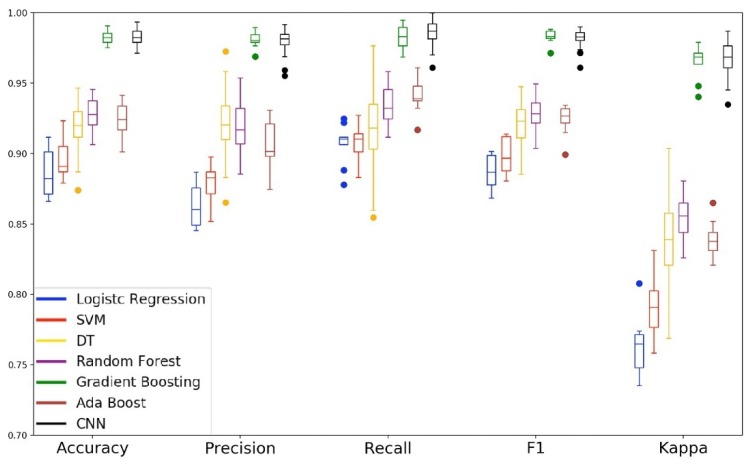
Performance of the models when analyzed with the data collected by the magnetometer.

**Figure 13 sensors-19-03174-f013:**
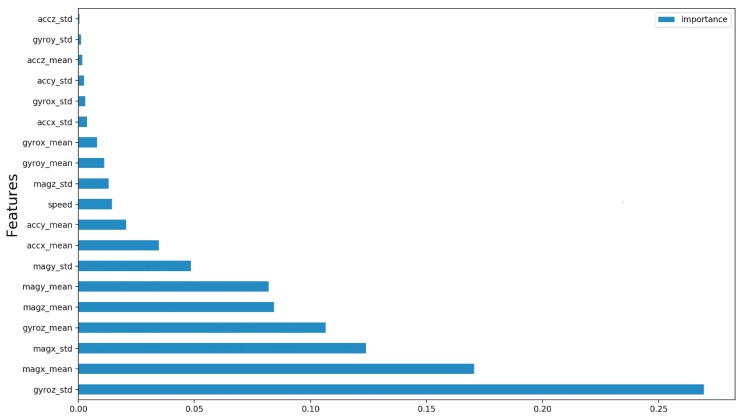
Feature importance computed by the Random Forest model.

**Figure 14 sensors-19-03174-f014:**
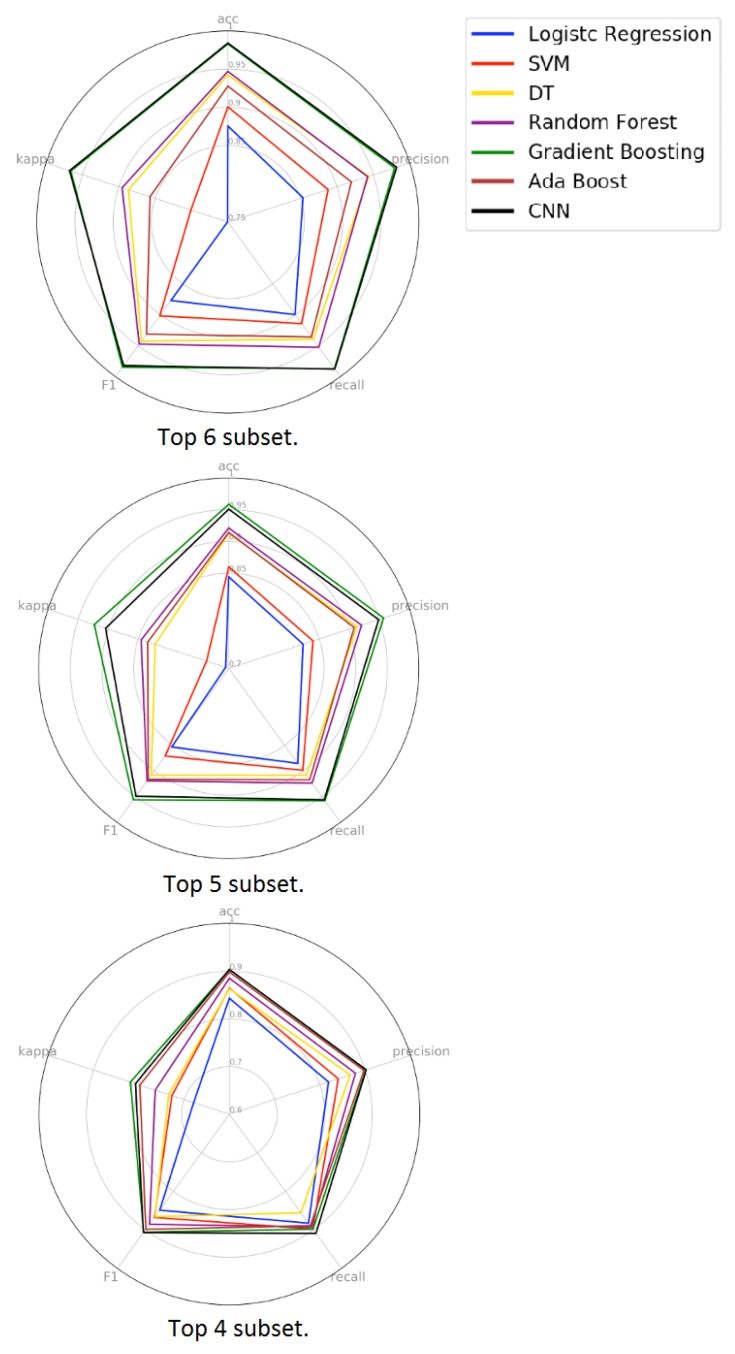
Radar chart with results from the third evaluation, considering the validation data. We can see that using the top 6 features, the overall results are better than using 5 or 4.

**Table 1 sensors-19-03174-t001:** Number of messages read by each volunteer. The subdivision of the driver or passenger column indicates that the volunteer has performed more than one trip with that profile.

Participant	Passenger	Driver
1	156	179
2	124	230
3	350	345
4	354	128
5	138	129
6	219	214
7	355	355
8	301	165 127
9	141	66 159
10	309	164 303
11	103	26 38 59
12	124	96 155 23
13	1975	452 354 380
14	152	30 56 81 50
15	87119	55 27 98 20
16	35136	31 170 144
17	16555	283 126
18	8121	182

**Table 2 sensors-19-03174-t002:** List of sensors used during data collection.

accelerometer-x	accelerometer-y	accelerometer-z
gyroscope-x	gyroscope-y	gyroscope-z
magnetometer-x	magnetometer-y	magnetometer-z
GPS-lat	GPS-long	

**Table 3 sensors-19-03174-t003:** List of extracted features from the collected dataset.

μ-acc-x	σ-acc-x	μ-acc-y	σ-acc-y
μ-acc-z	σ-acc-z	μ-gyro-x	σ-gyro-x
μ-gyro-y	σ-gyro-y	μ-gyro-z	σ-gyro-z
μ-mag-x	σ-mag-x	μ-mag-y	σ-mag-y
μ-mag-z	σ-mag-z	speed	

**Table 4 sensors-19-03174-t004:** List of machine-learning (ML) approaches filtered in the literature review. This table lists the number of works that have developed ML models in the context of driver distraction.

Models	Amount	References
*Support Vector Machine* (SVM)	12	[25,28,29,30,31,32,33,34,35,36,37,38]
*Logistic Regression*	8	[26,34,39,40,41,42,43,44]
*Artificial Neural Network* (ANN)	7	[31,36,45,46,47,48,49]
*K-Nearest Neighbor* (KNN)	4	[23,32,34,50]
*Convolutional Neural Network* (CNN)	2	[51,52]
*Adaptive Boosting* (AdaBoost)	2	[32,53]
*Decision Tree*	2	[41,54]
*Naive Bayesian*	2	[16,55]
*Recurrent Neural Networks* (RNN)	2	[24,36]
*Extreme Learning Machine* (ELM)	1	[35]

**Table 5 sensors-19-03174-t005:** Models’ grid search.

Model	C		Kernel	Gamma	
Support Vector Machine	prange (0.001, 1000.0, 10) **100.0**		**rbf** linear	prange (0.001, 1000.0, 10) **0.001**	
**Model**	**Criterion**	**Max depth**	**Max features**	**Min samples split**	
Decision Tree	Gini impurity **Entropy**	range (2, 40, 2) **10**	$\sqrt{n}$ $log_{2}\left(n\right)$	range (2, 40, 2) **16**	
**Model**	**N estimators**	**Criterion**	**Min sample split**	**Max depth**	**Max features**
Random Forest	range (75, 200, 25) **175**	**Gini impurity** Entropy	range (5, 100, 5) **80**	range (2, 10, 2) **6**	$\sqrt{n}$ $log_{2}\left(n\right)$
**Model**	**N estimators**			**Learning rate**	
Adaptive Boosting	range (75, 200, 25) **150**			range (0.1, 0.5, 0.1) **0.1**	
**Model**	**N estimators**	**Learning rate**		**Min sample split**	**Max depth**
Gradient Boosting	range (75, 200, 25) **100**	range (0.1, 0.5, 0.1) **0.4**		range (5, 100, 5) **80**	range (2, 10, 2) **4**

**Table 6 sensors-19-03174-t006:** Ω subsets used as input data in the second phase of the experiment.

**Subset 1: Accelerometer sensor**					
μ-acc-x	σ-acc-x	μ-acc-y	σ-acc-y	μ-acc-z	σ-acc-z
**Subset 2: Gyroscope sensor**					
μ-gyro-x	σ-gyro-x	μ-gyro-y	σ-gyro-y	μ-gyro-z	σ-gyro-z
**Subset 3: Magnetometer sensor**					
μ-mag-x	σ-mag-x	μ-mag-y	σ-mag-y	μ-mag-z	σ-mag-z

**Table 7 sensors-19-03174-t007:** Student’s hypothesis test of the models.

Model 1	Model 2	*p*-Value	H0
$GB_{1}$	$GB_{2}$	0.030	reject
$CNN_{1}$	$CNN_{2}$	0.576	accept
$CNN_{1}$	$GB_{1}$	0.123	accept
$CNN_{2}$	$GB_{2}$	0.327	accept
$CNN_{1}$	$GB_{2}$	0.841	accept
$CNN_{2}$	$GB_{1}$	0.215	accept

**Table 8 sensors-19-03174-t008:** Student’s hypothesis test of the models.

Model 1	Model 2	*p*-Value	H0
$CNN_{top6}$	$CNN_{1}$	0.092	accept
$CNN_{top6}$	$CNN_{2}$	0.154	accept
$CNN_{top6}$	$GB_{1}$	0.998	accept
$CNN_{top6}$	$GB_{2}$	0.012	reject
$GB_{top6}$	$GB_{1}$	0.075	accept
$GB_{top6}$	$GB_{2}$	0.476	accept
$GB_{top6}$	$CNN_{1}$	0.766	accept
$GB_{top6}$	$CNN_{2}$	0.675	accept
$GB_{top6}$	$CNN_{top6}$	0.032	reject

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
