# Peer review of "A Machine-Learning Approach to Distinguish Passengers and Drivers Reading While Driving"

_sensors, 2019, doi:10.3390/s19143174_

Round 1
Reviewer 1 Report
Although the content is of interest, authors must make a detailed review of the grammar and the writing of the entire document (sentence by sentence).
I recommend a review of the work by an English speaker to correct errors.
The wording should be revised because the text is very repetitive, and some sentences are very convoluted making the text difficult to follow.
Authors should explicitly clarify the number of text messages that volunteers read, both drivers and passengers. In the text it is mentioned: "eleven thousand simple were collected", but it is not clear if it refers to the number of text messages. Insofar as the average value and variance of the sensors, as well as the speed from the GPS, are calculated for each text message, the individual data samples should not be considered as data samples. The data samples are those used in the machine learning models (average values, variances and speed of each text message).
The machine learning models must be explained in detail, since the explanation of each of the models is partial and does not delve into all the important details.
A more exhaustive justification of the values of the hyperparameters in section "4. Setting up the hyperparameters" must be done, referring to references. Insofar as the models are known, the main novelty of the work is the application of use (detection of drivers from the sensors of smart phones) and the preliminary parameters of the models. Each hyperparameter must be explained in detail, so that section "3. Machine Learning Models" should be extensive enough to introduce such hyperparameters and their usefulness in the models.
The choice of the models used in the section "3. Machine Learning Models" has not been adequately justified.
Author Response
Thank you for your important contribution to this review process. We take all your recommendations into consideration and believe we have met all of them. In the new version of the manuscript, we highlight in bold the changes in the text.

Reviewer 2 Report
The authors describe a machine learning (ML) approach to differentiate between drivers and passengers who simply read text messages, without texting back. The ML algorithms draw on sensory data from the smartphones and nothing else. The authors collected these data from naturalistic driving of 18 volunteers who engaged in 11,000 reading instances, half as drivers and half as passengers. The results suggest that the magnetometer and gyroscope are the most discriminating sensors, while gradient boosting and convolutional neural networks the best ML algorithms for the job.
This work has merit and especially the ML part is well done. However, there are some issues with the paper, which the authors need to address. Specifically:
1) There are scant details about the experimental design and this is a major issue. For example, we do not know anything about data mix. Were all the 18 volunteers contributed both as passengers and drivers or some as passengers and other as drivers only? How many samples each of them produced? What was the duration of the study? Were the samples for each volunteer collected in a single driving session or in multiple sessions? If the latter how many and over how many days? What were the traffic and weather conditions? What were the directions of travel? All these need to be quantified, visualized, and described with statistics, because are consequential. For example, I am intrigued by the fact that magnetometer is the best performing sensor. This means either that the drivers hold the phone at a different orientation than passengers or that there is accidental bias in the data (e.g., coincidentally the driving samples were collected from a north facing highway while the passenger samples from an east facing highway).
2) The authors characterize imaging methods for detecting distractions as intrusive, while consider methods that depend on smartphone sensors as non-intrusive. I disagree with this statement. I do not believe that someone picking data from smartphone is performing a non-intrusive act either. In any case, it is a moot point. The authors contribute an additional method of identifying distractions and they should leave it at that without attaching negative labels to other methods. We need all the methods we can get and eventually market and cultural factors will determine which ones are used and which ones are not. Chances are that more than one methods will be used. Not to mention that in the upcoming Level 3 semi-automated vehicles there will be visual cameras in the interior of the vehicle as standard equipment. And, since the cameras will be there, they will be used anyway if not for distractions, for other purposes for sure. Not to mention that as we speak, smartphones are routinely unlocked via face recognition today (see iPhone X), and nobody is complaining for intrusiveness.
3) Important literature regarding distractions and ML approaches is missing. For example,
a study that abstracted distractions to sensorimotor, cognitive, and emotional types, while using multi-modal sensing channels to differentiate among them is notably absent.
Pavlidis, I., Dcosta, M., Taamneh, S., Manser, M., Ferris, T., Wunderlich, R., Akleman, E. and Tsiamyrtzis, P., 2016. Dissecting driver behaviors under cognitive, emotional, sensorimotor, and mixed stressors. Scientific Reports, 6, p.25651.
The full dataset for this study is publicly available and was described and analyzed in another paper:
Taamneh, S., Tsiamyrtzis, P., Dcosta, M., Buddharaju, P., Khatri, A., Manser, M., Ferris, T., Wunderlich, R. and Pavlidis, I., 2017. A multimodal dataset for various forms of distracted driving. Scientific Data, 4, p.170110.
There are many other papers that involve the use of physiological and driving data for detecting distractions. This entire line of literature is not mentioned at all by the authors. it offers some really cool alternatives, such as using heart rate data from smart watches as discriminating features for identifying distracted drivers. Relevant to this discussion, are absent rom the current paper recent ML methods that operate on multimodal data not only to detect, but also predict distracted states. For example:
Panagopoulos, G. and Pavlidis, I., 2019. Forecasting Markers of Habitual Driving Behaviors Associated With Crash Risk. IEEE Transactions on Intelligent Transportation Systems.
4) In Fig. 1 the car icon shows only a driver, but based on the design, it should also show a passenger, as the authors draw data from both roles.
5) The authors make the data available, which is very good. However, this is not the standard way for releasing datasets. The dataset needs to be available in a reputable public databank and needs to be accompanied by a data descriptor. See for an example here: https://osf.io/c42cn/
The authors are advised to act accordingly and include the link to the enhanced data site in their revised paper.
6) The authors choose to deal with the reading only (and not writing back) distractingly problem. I wonder how prevalent this problem is. I checked the literature but did not find any data about the prevalence of this problem. Anectodally, all the people I know and see always read and text back something. It would have been useful if the authors had a survey (at least on their volunteers), where they report how often they just read without texting back.
7) Very frankly, the size of the data the authors have is relatively small and ML is not absolutely needed. Good old statistics would probably have done an equally good job. The authors claim there is nonlinearity in the data, but they never provide quantitative evidence of that (e.g., a Q-Q plot or something). If the nonlinearity is structural (e.g., exponential distributions), then it can be corrected with a logarithmic transformation). In any case, the authors need to provide at the very least proof of nonlinearity.
8) Overall the paper is well written, There are some small style problems in the manuscript, though, the authors need to fix. For example, they refer to `eighteen volunteers'. Anything above 10 in a science paper needs to be stated in numerals and not words. So, it should be `18 volunteers and 11,000 samples.'
Author Response

(The authors gave the same response as above.)

Round 2
Reviewer 1 Report
The authors have responded appropriately to the comments of the reviewer. The manuscript is suitable for publication.
Author Response
We thank the reviewer for the comment. With your previous suggestions, we were able to develop an improved version of the paper. For this new revision, we performed a careful proofread aiming to remove any typo that remained in the manuscript.
Reviewer 2 Report
The authors responded to the majority of the points raised and the revised manuscript improved.
There are a couple of issues that I recommend be taken care of:
1) The authors state they submitted their data to the UCI repository and wait on UCI's response. They can also submit it elsewhere, where there is no wait all (e.g., figshare). Having the repository referenced in the paper will increase the value of their publication. If the repository gets published after the paper gets published, this reference will be missing adversely affecting the paper's impact. Also, the UCI repository is a small repository with questionable long-term support. I urge the authors to seek a `bigger' operation, which has higher chances to exist after 4-5 years.
2) The authors mention that they did literature search about ML methods for distracted driving in ACM and IEEE Xplore. These references are not in the Reference list. Since the authors made the effort to do this literature search, I wonder why they do not include the citations. This will add about 40 more references to the manuscript and will better reflect the work the authors did to prepare their ML approach. I also mentioned this in my original review: There is little (if any) citation of ML methods in distracted driving (which is half the paper here) - just the number of articles the authors found, without really revealing them.
Author Response
We thank the reviewer for the comments. With your previous and current suggestions, we were able to develop an improved version of the paper. For this new revision, we performed a careful proofread aiming to remove any typo that remained in the manuscript.
Related to: (1) “The authors state they submitted their data to the UCI repository and wait on UCI's response. They can also submit it elsewhere, where there is no wait all (e.g., figshare). Having the repository referenced in the paper will increase the value of their publication. If the repository gets published after the paper gets published, this reference will be missing adversely affecting the paper's impact. Also, the UCI repository is a small repository with questionable long-term support. I urge the authors to seek a `bigger' operation, which has higher chances to exist after 4-5 years.”
Response: We acknowledge your suggestion and we uploaded the dataset at figshare, as mentioned in the new version of the manuscript. The database available at https://figshare.com/articles/Passengers_and_Drivers_reading_while_driving/8313620
Related to (2) “The authors mention that they did literature search about ML methods for distracted driving in ACM and IEEE Xplore. These references are not in the Reference list. Since the authors made the effort to do this literature search, I wonder why they do not include the citations. This will add about 40 more references to the manuscript and will better reflect the work the authors did to prepare their ML approach. I also mentioned this in my original review: There is little (if any) citation of ML methods in distracted driving (which is half the paper here) - just the number of articles the authors found, without really revealing them.”
Response: We added the references to the paper, they can be seen in Table 4.